# Dynamic Causal Influence Learning in Cooperative Multi-Agent Reinforcement Learning

## Abstract

The credit assignment problem remains a fundamental challenge in multi-agent reinforcement learning (MARL) due to the complex environment dynamics. In this paper, we define *A-Q influence* to capture the state-dependent causal influence relationship between individual actions and individual action value functions in an MARL problem. Then influence-based local value functions (ILVFs) are constructed and shown to be equivalent to the global value function in terms of policy gradient estimation. To efficiently attain the agent-wise A-Q influence, we propose to infer A-Q influence according to *state influence*, which is learned by a Gumbel-max attention mechanism. To evaluate the effectiveness of ILVF, we integrate it into the MAPPO framework and propose the ILVF-P algorithm. Extensive experiments on diverse MARL benchmarks reveal that ILVF-P consistently surpasses strong baselines, underscoring its benefits in facilitating the training efficiency.

## 1 Introduction

Multi-agent reinforcement learning (MARL) has garnered increasing attention in recent years due to its remarkable success in real-world applications such as traffic signal control Wu et al. (2020); Agrahari et al. (2024), autonomous vehicle coordinationKiran et al. (2021); Chen et al. (2024) and robotic control Wang et al. (2023a); Tang et al. (2025). Nevertheless, existing approaches still suffer from limited scalability because of the non-stationarity of the environment. More specifically, the action of one agent always has influence on other agents, and the inter-agent causal influence changes all the time.

As a fundamental paradigm in MARL, independent learning (e.g. IQL Tan (1993); Jiang & Lu (2022; 2023) and IPPO De Witt et al. (2020)) exhibits advantages in scalability and competitive performance in cooperative tasks. However, treating other agents as environmental components introduces non-stationarity and makes convergence more difficult. To address this, the Centralized Training with Decentralized Execution (CTDE) framework is widely adopted, which leverages global information during training while executing based on local observations. Although CTDE theoretically aligns policies via global value functions, practical implementation struggles to quantify individual contributions to collective goals. This exacerbates the difficulty of credit assignment, a critical challenge that intensifies with increasing system scale Phan et al. (2021). While shared reward settings have received increasing attention, the credit assignment problem in reward-sum settings Jiang et al. (2024); Albrecht et al. (2024) remains significant and continues to face intractable challenges.

The credit assignment problem arises from the difficulty of accurately evaluating each agent's true contribution to the overall objective. In this work, we define the *A-Q influence* to capture the state-dependent influence relationship between individual actions and individual action-value functions. Identifying the A-Q influence relationships among agents essentially constitutes a process of credit assignment. Inspired by the philosophical notion of *causal responsibility*—an agent should be only accountable for outcomes it causally influences, we construct an influence-based local value function (ILVF) for each agent, by aggregating the state-value functions of those agents it has A-Q influence on. We theoretically prove that each individual ILVF is equivalent to the global value function in terms of policy gradient, therefore can be used to guide policy updates.

However, the A-Q influence relationships among agents are dynamic and difficult to obtain through analytical methods. To overcome this challenge, we propose to infer it based on *state influence*, which describes the influence relationship of each individual state on the value functions and actions of other agents. A Gumbel attention mechanism is proposed to learn the state influence. Experimental results demonstrate that inferring A-Q influence through the learned state influence is more effective than learning A-Q influence directly.

Our main contributions are as follows. (i) We define a state-dependent local value function, termed ILVF, for each agent based on the A-Q influence and prove the equivalence of the proposed ILVF and the global value function in terms of policy gradient. (ii) We propose a Gumbel attention mechanism to learn inter-agent state influence as part of the critic network training; (iii) Based on inter-agent state influence, we infer A-Q influence and construct ILVF to guide policy optimization. (iv) By conducting extensive numerical experiments, it is shown that the ILVF approach significantly facilitates policy learning and cooperation among agents, especially when large-scale problems are encountered.

## 2 RELATED WORK

### 2.1 VALUE FUNCTION FACTORIZATION

Value Function Factorization (VFF) Rashid et al. (2020b); Zhou et al. (2023); Li et al. (2024); Su et al. (2021) under the CTDE framework Foerster et al. (2018) decomposes the global action-value function into local utilities to enhance decentralized policy learning. While VDN Sunehag et al. (2017) enforces linear decomposition with limited representational capacity, QMIX Rashid et al. (2020b) employs state-conditioned nonlinear mixing networks to implement monotonic decomposition. Subsequent innovations Rashid et al. (2020a); Son et al. (2019); Phan et al. (2021); Li et al. (2024) further expand function representation through alternative factorization paradigms. These methods collectively face critical structural constraints as the compounding approximation errors in value decomposition severely degrade training signal propagation, particularly in large-scale systems. Several studies Wang et al. (2020); Zhang et al. (2021); Zhou et al. (2023) extend value decomposition to policy-based actor-critic frameworks, thereby enhancing training efficiency and improving credit assignment. Nevertheless, the above-mentioned approaches still suffer from low scalability since they never consider the underlying inter-agent coupling relationship.

### 2.2 LOCAL VALUE FUNCTIONS IN NETWORKED SYSTEMS

Indeed, structural properties of networked systems have been exploited in Qu et al. (2020); Zohar et al. (2022); Ma et al. (2024); Jing et al. (2024) to construct local value functions, based on which scalability and credit assignment issues are better addressed. Qu et al. (2020) adopts $\kappa$-hop truncation on the network to enhance scalability, but it relies on assumptions of state-transition and reward independence. Ma et al. (2024) relaxes the assumptions in Qu et al. (2020), yet the neighborhood is defined solely by observation relations, which leads to biased gradients. Jing et al. (2024) derives local value functions that are equivalent to global value function in policy gradients, thereby improving optimality in sparse settings, but it struggles in dense coupled scenarios. All of these methods assume that the system topology is static and known in advance. However, many real-world systems exhibit dynamic and unknown topologies, making such methods unsuitable in practice.

### 2.3 CAUSAL INFERENCE IN MARL

While causal inference has been extensively explored in single-agent reinforcement learning (RL), its application in multi-agent settings remains relatively underexplored Grimbly et al. (2021). Existing works predominantly focus on leveraging learned causal structures to enhance credit assignment Seitzer et al. (2021); Du et al. (2024); Wang et al. (2023b) or generalization across environments Mutti et al. (2023b;a). Seitzer et al. (2021); Du et al. (2024) use conditional mutual information to define causal influence, enabling agents to identify controllable components in the state transition process. Wang et al. (2023b) further incorporates individual rewards into consideration and employs Dynamic Bayesian Networks to infer the causal relationships, thereby facilitating interpretable credit assignment through reward decomposition. These methods consider only the instantaneous influence among variables and, therefore, fail to quantify the contribution of such influence to the overall objective.

# 3 BACKGROUND

## 3.1 PRELIMINARIES

The considered Markov Decision Process is represented by a tuple $< \mathcal{N}, \mathcal{S}, \mathcal{A}, R, P, O, \gamma >$, where $\mathcal{N} = \{1, ..., n\}$ denotes the set of $n$ agents, $s = (s_1, ..., s_n) \in \mathcal{S}$ represents the global environment state, where $s_i$ denotes the individual state of each agent; $a = (a_1, ..., a_n) \in \mathcal{A}$ denotes the joint action of all agents, formed by concatenating individual actions $a_i \in \mathcal{A}_i$, $\mathcal{A}_i$ is a finite discrete action space; $P : \mathcal{S} \times \mathcal{A} \to \mathcal{S}$ represents the state transition function; $R = \sum_{i=1}^{n} R_i(s, a) : \mathcal{S} \times \mathcal{A} \to \mathbb{R}$ is the global reward function[1], $R_{s,a}$ is the individual reward for agent $i \in \mathcal{N}$; $O_i : \mathcal{S} \to \prod_{j \in N_i^o(s) \cup \{i\}} \mathcal{S}_j$ denotes the observation function for agent $i$, here $N_i^o(s) \subset \mathcal{N} \setminus i$ is the neighbor set of agent $i$; $\pi(a|s) : \mathcal{S} \times \mathcal{A} \to [0, 1]$ represents the joint policy decomposed as $\pi(a|s) = \prod_{i=1}^{n} \pi_i(a_i|o_i)$, where $\pi_i(a_i|o_i) \to [0, 1]$ is the individual policy based on local observations. The initial states are sampled from a distribution $\rho : \mathcal{S} \to [0, 1]$; $\gamma \in (0, 1)$ denotes the discount factor.

The individual state-value function for agent $i$ is defined as $V_i^\pi(s) = \mathbb{E}_\pi \left[ \sum_{t=0}^{T} \gamma^t R_i(s^t, a^t) \Big| s^0 = s \right]$. The individual action-value function and advantage function for agent $i$ are $Q_i^\pi(s, a) = \mathbb{E}_\pi \left[ \sum_{t=0}^{T} \gamma^t R_i(s^t, a^t) \Big| s^0 = s, a^0 = a \right]$ and $A_i^\pi(s, a) = Q_i^\pi(s, a) - V_i^\pi(s)$, respectively. The global state-value function is defined as $V^\pi(s) = \sum_{i=1}^{n} V_i^\pi(s)$ and the corresponding global action-value function and advantage function are $Q^\pi(s, a) = \sum_{i=1}^{n} Q_i^\pi(s, a)$ and $A^\pi(s, a) = Q^\pi(s, a) - V^\pi(s)$, respectively. The problem of interest is to learn an optimal policy $\pi_i(a_i|o_i)$ for each agent $i \in \mathcal{N}$, such that the following objective is maximized:

$$J(\theta) = \mathbb{E}_{s \sim p^\pi}[V^\pi(s)], \tag{1}$$

where $p^\pi$ is the state distribution.

In our formulation, although both the transition and reward functions depend on the global state and joint actions, practical dependencies often reduce to state-specific and agent-level interactions. Meanwhile, the observation function captures how local perceptions and communication channels convey individual state information. Together, these structured dependencies enable effective analysis and learning of inter-agent influence, despite partial observability and complex environmental couplings.

## 3.2 ATTENTION MECHANISM AND GUMBEL-SOFTMAX TRICK

In this subsection, we introduce the attention mechanism and Gumbel-Softmax Trick, which provide foundations for our MARL framework.

**Attention Mechanism**. The attention mechanism (AM), a cornerstone of deep learning Ji et al. (2023), emulates human perceptual focus by dynamically allocating computational resources: high-resolution local perception with low-resolution global awareness via weighted feature allocation. This universal architecture mitigates irrelevant information interference and has been widely adopted across Natural Language Processing (NLP), Computer Vision (CV), and speech recognition Hu et al. (2023). A general form of the attention mechanism is formulated as:

$$\text{Attention} = f(g(x), x), \tag{2}$$

where $g(x)$ represents the generation of attention, which corresponds to the process of attending to the discriminative regions. $f(g(x), x)$ means processing input $x$ based on the attention $g(x)$, which is consistent with processing critical regions and obtaining information Guo et al. (2022). We take self-attention Wang et al. (2018) and squeeze-and-excitation (SE) attention Hu et al. (2018) as examples. For self-attention, $g(x)$ and $f(g(x), x)$ can be written as $g(x) = \text{Softmax}(QK)$ and $f(g(x), x) = g(x)V$, where $Q, K, V = \text{Linear}(x)$. For SE, $g(x)$ and $f(g(x), x)$ can be written as $g(x) = \text{Sigmoid}(\text{MLP}(\text{GAP}(x)))$ and $f(g(x), x) = g(x)x$.

**Gumbel-Softmax Trick.** The Gumbel-Softmax trick provides a differentiable surrogate for discrete sampling, vital in latent variable modeling and reinforcement learning. It injects Gumbel noise

---

[1]Our method can naturally extend to scenarios without a predefined correspondence between agents and reward functions.

$g_i = -\log(-\log u_i)$, where $u_i$ is sampled from the uniform distribution $U(0,1)$, and approximates $\arg\max$ through temperature-controlled softmax:

$$\tilde{y}_i = \frac{\exp[(\log \pi_i + g_i)/\tau]}{\sum_{j=1}^{k} \exp[(\log \pi_j + g_j)/\tau]}, \tag{3}$$

where $\pi$ is a k-ary discrete distribution with $\pi_i$ denoting the probability of category $i$. As $\tau \to 0$, $\tilde{\boldsymbol{y}} = [\tilde{y}_1, \ldots, \tilde{y}_k]$ converges to one-hot encoding while maintaining differentiability. This trick enables gradient-based optimization for discrete decisions in VAEs Jang et al. (2016), RL policies Metz et al. (2017), and graph sampling Kipf & Welling (2016). Recent extensions like adaptive annealing Campbell et al. (2022) and entropy constraints Vahdat et al. (2018) refine its exploration-exploitation balance.

# 4 METHOD

In this section, we present our main methodology based on A-Q influence and the influence-based local value function (ILVF).

## 4.1 A-Q INFLUENCE AND INFLUENCE-BASED LOCAL VALUE FUNCTIONS

Causal graph models (CGMs) have been extensively utilized in existing studies, e.g., Du et al. (2024); Seitzer et al. (2021), to characterize influence among agents' states. These works primarily focus on how one agent's decision affects another's state transition, without comprehensively evaluating the contribution of these decisions to the overall task objective. In contrast, we define A-Q influence as a measurement that comprehensively captures various forms of inter-agent coupling and reflects whether an agent's decision substantially contributes to the task performance, which is described in Figure 1.

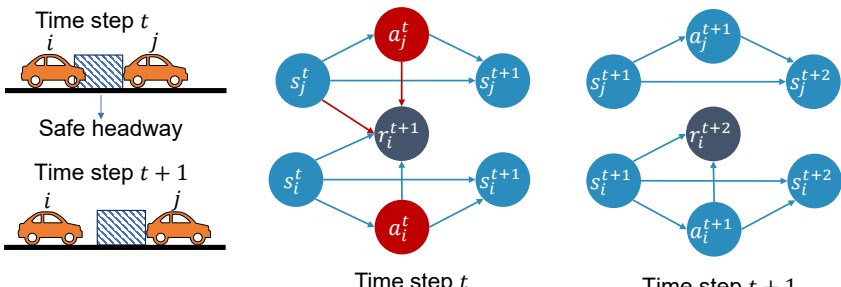

Figure 1: Causal Graphical Models (CGMs) in Cooperative Adaptive Cruise Control (CACC ). A node corresponds to a random variable and a directed edge from $x$ to $y$ indicates that $x$ has a causal influence on $y$. It illustrates how inter-agent influence relationships vary across different states and presents the corresponding CGM diagrams at each time step. The state of each vehicle includes its position, speed, and headway, while the action corresponds to acceleration. The objective is to achieve the desired headway and speed.

We now define the A-Q influence and the influence-based local value function.

**Definition 1** (A-Q Influence). *Given policy $\pi$ and state $s$, if there exists $a'_j \in \mathcal{A}_j$ such that:*

$$Q_i^\pi(s, a_j, a_{\mathcal{N} \setminus j}) \neq Q_i^\pi(s, a'_j, a_{\mathcal{N} \setminus j}), \tag{4}$$

*then agent $j$ is considered to have* A-Q influence *on agent $i$ under policy $\pi$ and state $s$.*

We draw inspiration from the principle of causal responsibility, which resonates with Hans Jonas's ethics of responsibility Chockler & Halpern (2004) and emphasizes that a moral subject should be held accountable only for the results it causally influences. In the context of MARL, this implies that a local value function can be constructed, which reflects the agent's actual sphere of influence, rather than attributing global outcomes indiscriminately.

Define $N_i(\pi, s) = \{j | j \in \mathcal{N}, j \text{ satisfies (4)}\}$, which denotes the set of agents that exhibit A-Q influence on agent $i$ in state $s$, and $N_i^+(\pi, s) := \{j \mid i \in N_j(\pi, s) \text{ for all } j \in \mathcal{N}\} \cup \{i\}$, which denotes the set of agents whose individual action-value function is influenced by the action of agent $i$ in state $s$. In practice, it may be difficult to obtain the exact $N_i^+(\pi, s)$. Thus, we focus on learning a more general set $N_i^l(\pi, s)$, which contains $N_i^+(\pi, s)$ as a subset, i.e., $N_i^+(\pi, s) \subseteq N_i^l(\pi, s) \subseteq \mathcal{N}$.

**Definition 2** (Influence-Based Local Value Function). *The sum of individual state-value functions of agents in the set $N_i^l(\pi, s)$:*

$$\hat{V}_i^\pi(s) = \sum_{j \in N_i^l(\pi, s)} V_j^\pi(s), \tag{5}$$

*is referred to as an influence-based local value function (ILVF) for agent $i$.*

When $N_i^l(\pi, s) = \mathcal{N}$, the ILVF $\hat{V}_i^\pi(s)$ becomes the global state-value function. When $N_i^l(\pi, s) = N_i^+(\pi, s)$, it is called the minimal ILVF. Similarly, we can define the corresponding local action-value function $\hat{Q}_i^\pi(s, a)$ and local advantage function $\hat{A}_i^\pi(s, a)$. The local value function constructed based on A-Q influence not only facilitates credit assignment and embodies philosophical principles but also enjoys the following theoretical guarantee.

**Theorem 1.** *[Policy Gradient Equivalence] The gradient of the ILVF $\hat{V}_i^\pi(s)$ in (5), with respect to policy $\pi_i(a_i|o_i, \theta_i)$ is equivalent to that of the global value function $V^\pi(s)$.*

The equivalence established in Theorem 1 constitutes a key distinction from prior approaches Du et al. (2024); Seitzer et al. (2021); Wang et al. (2019) that capture inter-agent influence via intrinsic rewards. The proof of Theorem 1 is postponed to Appendix A. A *sparse* local value function[2] not only facilitates effective credit assignment but also improves sample efficiency significantly.

A straightforward approach to constructing local value functions is to directly learn the A-Q influence between agents at each state while fitting the individual action-value functions with neural networks. However, in the scenarios when the reward mainly depends on states with an explicit state-reward formulation, the influence of actions on the reward is mediated by environment dynamics. As a result, learning A-Q influence with neural networks is more sophisticated and entails higher sample complexity than learning state influence, which is experimentally verified in ablation study 5.3. Therefore, we instead choose an alternative strategy—**inferring A-Q influence via state influence**.

### 4.2 INFERRING A-Q INFLUENCE

In Figure 1, the state $s_j^t$ can directly affect $r_i^{t+1}$ under reward coupling, or indirectly influence the future rewards of agent $i$ by affecting $a_i^{t+1}$, and the state transitions of agent $f$. These three types of influence collectively represent the impact of state $s_j$ on value functions $V_i^\pi(s)$ and $Q_i^\pi(s, a)$, which we define as *S-V influence*.

**Definition 3** (S-V Influence). *Given policy $\pi(a|s)$ and state $s$, if there exists $s_j'$ such that $V_i^\pi(s_j, s_{\mathcal{N}\backslash j}) \neq V_i^\pi(s_j', s_{\mathcal{N}\backslash j})$ or $Q_i^\pi(s_j, s_{\mathcal{N}\backslash j}, a) \neq Q_i^\pi(s_j', s_{\mathcal{N}\backslash j}, a)$ for some $a \in \mathcal{A}$, then agent $j$ is said to have S-V influence on agent $i$ under policy $\pi$ and state $s$.*

Let $[x]_{x \in X}$ denote a vector with components being the elements in $X$. Next we further define *S-P influence* to describe the relationship between two agents where an individual state has a sufficiently strong influence on another individual policy.

**Definition 4** (S-P Influence). *Given policy $\pi(a \mid s)$ and state $s$, define*

$$\Pi_i^j(\pi, s) = \text{span}\left\{\left[\pi_i(a_i \mid s) - \pi_i(a_i \mid s_{\mathcal{N}\backslash j}, s_j')\right]_{a \in \mathcal{A}_i}, s_j' \in \mathcal{S}_j\right\}. \tag{6}$$

*If $\dim(\Pi_i^j(\pi, s)) = |\mathcal{A}_j| - 1$, then agent $j$ is said to have S-P influence on agent $i$ under policy $\pi$ and state $s$.*

The dimension of the space $\Pi_i^j$ reflects the number of independent directions in which the policy $\pi_i$ can be altered by varying $s_j$. According to Lemma 1 in Appendix, the S-P influence of agent $j$

---

[2]A local value function is considered to be sparser when it aggregates fewer individual value functions, and denser otherwise.

on agent $i$ implies that $\Pi_i^j$ has a maximum dimension. In this case, we are able to establish a clear connection between state influence (S-P and S-V) and A-Q influence.

Let $N_i^V(\pi, s), N_i^P(\pi, s) \subseteq \mathcal{N}$ denote the sets of agents exerting S-V and S-P influence on agent $i$ under state $s$ and policy $\pi$, respectively. Next, we propose a theorem to show how to infer the A-Q influence based on the state influence. The proof is given in Appendix A.

**Theorem 2.** *[A-Q Influence Elimination] Given policy $\pi(a \mid s)$ and state $s$, the agents in $I_i(\pi, s) = \{j | (\mathcal{N} \setminus N_i^V(\pi, s)) \cap N_j^P(\pi, s) \neq \emptyset, j \in \mathcal{N}\}$ have no A-Q influence on the agent $i$.*

Throughout this paper, we make the following assumption on S-P influence:

**Assumption 1.** *Every agent has S-P influence on itself for any policy and state.*

This assumption is valid since in most environments where an agent's decision is strongly influenced by its own state Seraj et al. (2022); Chu et al. (2020); Lowe et al. (2017); Papoudakis et al. (2021). Let $\hat{I}_i(\pi, s) = \mathcal{N} \setminus N_i^V(\pi, s)$. Under Assumption 1, we have $\hat{I}_i(\pi, s) \subset I_i(\pi, s)$ and $N_i(\pi, a) \subseteq N_i^V(\pi, s)$, i.e., if $j \notin N_i^V(\pi, s)$, agent $j$ does not have A-Q influence on agent $i$ under state $s$. Thus, we can determine sparse local value functions for the agents based on the S-V influences between the agents.

## 4.3 LEARNING STATE INFLUENCE FOR AGENT COLLABORATION

In this subsection, we propose a Gumbel attention mechanism to learn state influence while training individual state-value and action-value functions, and construct local advantage functions to guide policy optimization. The training architecture is shown in Figure 2.

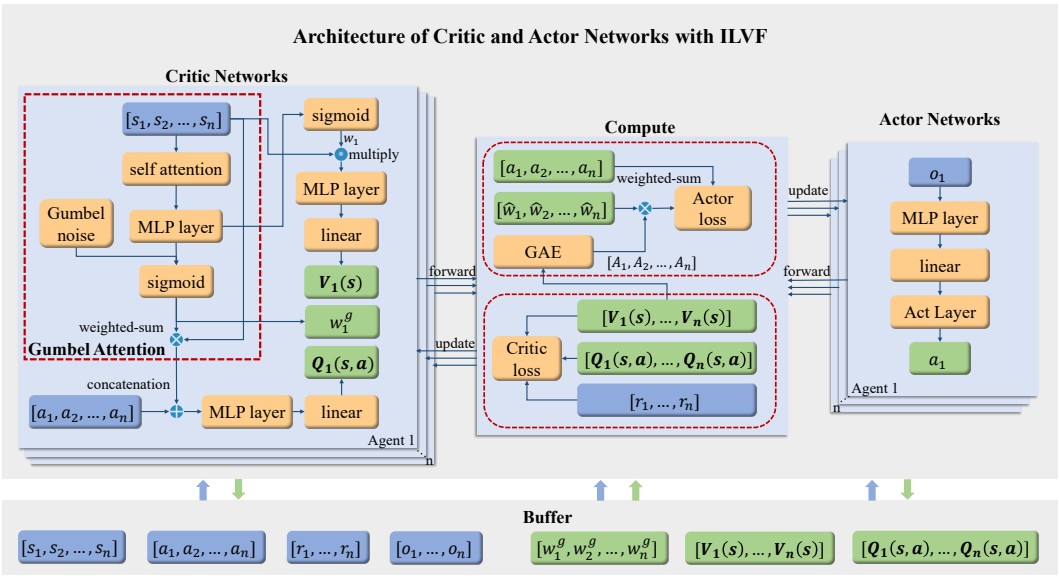

Figure 2: The learning architecture with ILVF. The blue, orange, and green boxes represent external inputs, modules within the network, and intermediate computed variables, respectively. The Gumbel attention mechanism is integrated into the critic network to learn the agent-wise A-Q influence relationships, represented by the matrix $W$, which is then used to construct local advantage functions to facilitate policy learning for individual agents.

### 4.3.1 GUMBEL ATTENTION MECHANISM FOR STATE INFLUENCE LEARNING

The Gumbel Attention Mechanism introduces Gumbel noise into the attention weights to induce near-discrete representations, thereby enabling the identification of state influence relationships among agents in a differentiable and learnable manner. The samples of the Gumbel distribution are generated by the transformation $g = -\log(-\log(u)), u \sim U(0, 1)$.

The state-value function $V_i^\pi(s)$ and action-value function $Q_i^\pi(s, a)$ are parameterized with neural networks and represented by $V_{\Phi_i^V}^\pi$ and $Q_{\Phi_i^Q}^\pi(s, a)$, respectively:

$$V_{\Phi_i^V}^\pi(s) = f_{\phi_i^V}(w_{i1}s_1, \ldots, w_{in}s_n), \quad Q_{\Phi_i^Q}^\pi(s, a) = f_{\phi_i^Q}\left(\sum_{j=1}^n w_{ij}^g s_j, a\right), \tag{7}$$

where $w_{ij}, w_{ij}^g \in [0, 1], \forall i, j \in \{1, \ldots, n\}$, are attention weights, with $w_i(s) = [w_{i1}, \ldots, w_{in}]^\top$ and $w_i^g(s) = [w_{i1}^g, \ldots, w_{in}^g]^\top$ denoting the attention vectors for agent $i$. Let $h_i = f_{\phi_i^h}(s) = [h_{i1}, \ldots, h_{in}] \in \mathbb{R}^n, i = 1, \ldots, n$ represent the output vector of the neural network $f_{\phi_i^h}(\cdot)$. To handle heterogeneous action spaces, we pad each agent's action representation to a unified dimension before applying the weighting and aggregation. The calculation of the attention weights is as follows:

$$w_{ij} = \text{sigmoid}(h_{ij}) = (1 + e^{-h_{ij}})^{-1}, \quad w_{ij}^g = \text{sigmoid}(h_{ij} + g_{ij}), \tag{8}$$

where $g_{ij}$ represents a vector of Gumbel noise, $\phi_i = [\phi_i^h, \phi_i^V, \phi_i^Q]$ represents the parameter set of the critic network of agent $i$. The attention weights $w_{ij}$ and $w_{ij}^g$ are used for aggregating state information in the individual state-value and action-value functions, respectively. Thus, after learning, $h_i$ is expected to encapsulate the information of S-V influences.

The method of state aggregation affects both value function fitting and attention weight learning. Weighted summation of states helps to focus on relevant information for value estimation, improving attention weight learning, but compressing state features and increasing approximation error. In contrast, concatenating weighted states retains full state inputs and enhances value fitting, yet fails to yield true attention coefficients due to entanglement with other network parameters. Introducing Gumbel noise promotes discrete attention weights that better reflect inter-agent influence, though it exacerbates value approximation error.

To balance these trade-offs, we apply noisy attention weights $w_{ij}^g$ in the action-value function network to perform weighted summation over agent states, ensuring accurate learning of attention coefficients. On the other hand, in the state-value function network, we concatenate states weighted by attention weights $w_{ij}$ to retain the full state input and enhance value approximation accuracy. During policy optimization, we employ Generalized Advantage Estimation (GAE) based on the state-value function to estimate the advantage function. The loss function of the critic network is defined as:

$$\mathcal{L}(\phi_i) = \sum_{t=0}^T \left[(G_i^t - V_{\phi_i}^\pi(s^t))^2 + (G_i^t - Q_{\phi_i}^\pi(s^t, a^t))^2\right], \tag{9}$$

where $G_i^t = \sum_{k=t}^T \gamma^{k-t} r_i^k$ denotes the discounted return for agent $i$ at time $t$. After training each critic network, the state-dependent attention weights with Gumbel noise $w_{ij}^g \in [0, 1], \forall i, j \in \{1, \ldots, n\}$, are learned and collectively form the attention matrix $W^g(s)$, which captures the overall inter-agent S-V influence.

### 4.3.2 STATE-SPECIFIC LOCAL ADVANTAGE FUNCTION

We use GAE Schulman et al. (2015) to estimate individual advantage functions $A_{i,t}^{GAE(\gamma,\lambda)}$, where $\delta_{i,t}^V = r_i^t + \gamma V_i^\pi(s^{t+1}) - V_i^\pi(s^t)$. The influence-based local advantage function for agent $i$ can be calculated by:

$$\hat{A}_i^\pi(s^t, a^t) = \sum_{j=1}^n \hat{w}_{ij}(s^t) A_{j,t}^{GAE(\gamma,\lambda)}, \tag{10}$$

where $\hat{w}_{ij}$ denotes the $(i, j)$-th entry of the influence matrix $\hat{W}$, which is obtained by sampling from a set of Bernoulli distributions whose parameters are given by the corresponding entries of $W^g$. The policy gradient of agent $i$ is obtained by:

$$\nabla_{\theta_i} J(\theta_i) = \mathbb{E}_{s \sim p(.|s,a), a \sim \pi(.|s,\theta)} \left[\nabla_{\theta_i} \log \pi_i(a_i|s, \theta) \hat{A}_i^\pi(s, a)\right], \tag{11}$$

according to the stochastic policy gradient theorem Sutton et al. (1998).

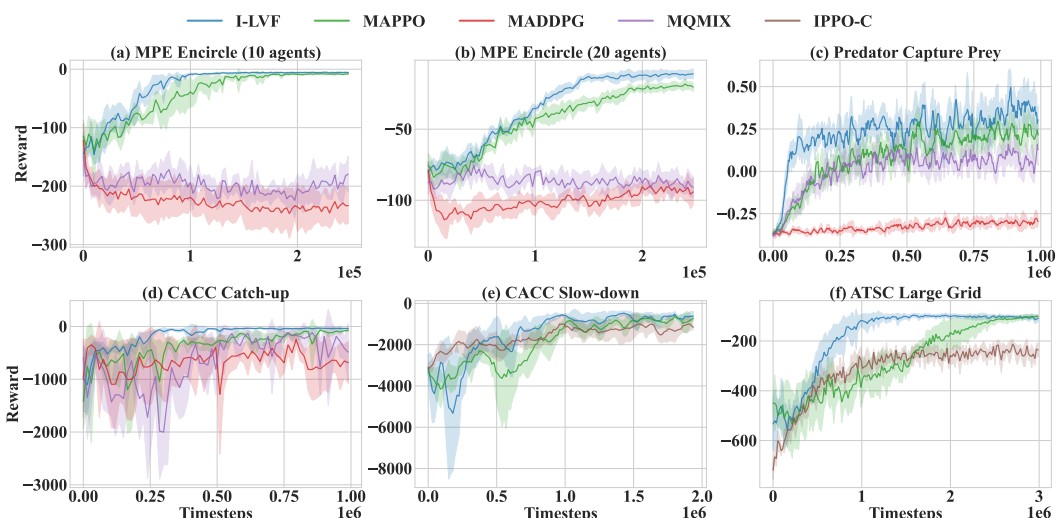

Figure 3: Reward convergence comparison in various environments.

## 5 EXPERIMENTS

### 5.1 ENVIRONMENTS AND BASELINES

To validate the effectiveness of ILVF in improving credit assignment and sample efficiency under reward sum settings, we integrate ILVF into the MAPPO framework, denoted by ILVF-P, and conduct extensive experiments in classical multi-agent cooperation environments, including MPE Encircle Agarwal et al. (2019), Predator Capture Prey (PCP) Seraj et al. (2022), CACC Chu et al. (2020), and ATSC Chu et al. (2020). In the comparisons with state-of-the-art MARL algorithms such as MAPPO Yu et al. (2022), IPPO De Witt et al. (2020), QMIX Rashid et al. (2020b), and MADDPG Lowe et al. (2017), we demonstrate the superior performance of the proposed ILVF construction and learning method. Comprehensive implementation and configuration details of these environments and baseline methods are provided in Appendix B.

### 5.2 COMPARISON RESULTS

The reward convergence curves of the proposed ILVF-P algorithm and the baseline algorithms in the PCP, MPE Encircle, CACC, and ATSC environments are shown in Figure 3. Across all experimental scenarios, our method consistently outperforms the baselines. In the early stages of training, ILVF-P achieves faster convergence, demonstrating the benefit of learning the state influence in accelerating policy optimization. In the later stages, ILVF-P achieves global average rewards that are comparable to or better than those of the baselines.

In the MPE Encircle environment, a comparison between Figure 3(a) and (b) reveals that as the problem scales up, ILVF-P achieves faster convergence and better overall performance. This indicates the scalability and effectiveness of the proposed method. As shown in Figure 3(c), the results in the PCP environment confirm the effectiveness of ILVF-P in heterogeneous multi-agent coordination tasks. We attribute the observed performance gap, also illustrated in Figure 3(b), to the clipping mechanism used in MAPPO to stabilize the training, which may introduce bias into the policy gradients. By contrast, ILVF effectively mitigates this cumulative bias, leading to improved performance.

In the more complex CACC slow-down and ATSC large-grid scenarios, MADDPG and QMIX are found to be highly inefficient and time-consuming. Therefore, we select MAPPO and IPPO-C as baseline algorithms for comparison. MAPPO and IPPO-C can be viewed as two representative extremes in local value function design: MAPPO corresponds to the case where the value function is global, while IPPO-C corresponds to the case where it is fully individual. Positioning ILVF-P between these two extremes allows us to examine whether ILVF provides benefits beyond global or individual value functions.

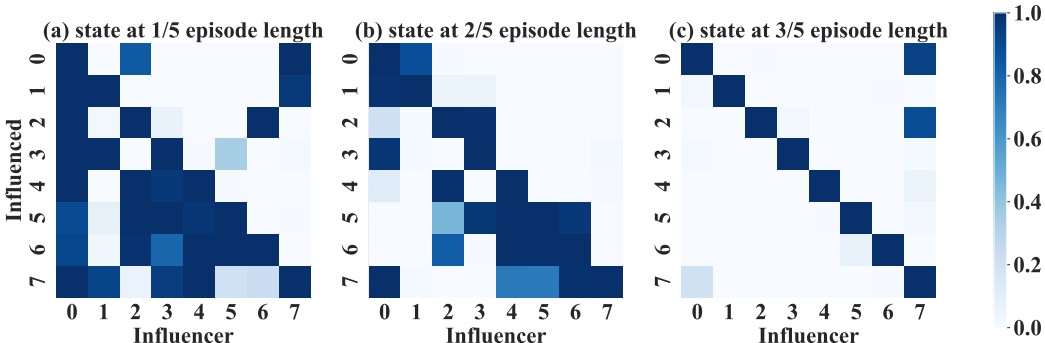

Figure 4: The heatmap of $\hat{W}$ in different state.

Next, we demonstrate how the influence matrix $\hat{W}$ evolves during the learning process, by focusing on the CACC Slowdown scenario in different states within a single episode. As shown in Figure 4, at 1/5 of the episode, matrix $\hat{W}$ is relatively dense, as the episode progresses, the influence relationships gradually become sparser. This can be explained as follows. In the early stage, none of the agents reach their desired speed or headway, and the trailing vehicles are significantly influenced by the leading ones, resulting in a $\hat{W}$ with an approximate lower-triangular structure. In the middle stage, as inter-vehicle spacing is adjusted, each agent primarily influences only a few neighboring agents. In the final stage, each agent reaches target speed and headway, the influence relationships diminish, and $\hat{W}$ becomes nearly diagonal. In conclusion, the learned influence matrix well describes the dynamic inter-agent influence, therefore significantly enhances the learning efficiency.

### 5.3 THE MOTIVATION FOR INFERRING A-Q INFLUENCE

We further conduct experiments to examine the rationale for learning state influence to inferring A-Q influence. To this end, we design a baseline variant that integrates the Gumbel attention mechanism into the action-value function network to learn the A-Q influence among agents directly. We refer to this method as ILVF-P (no inferring), and its architecture is given in Figure 7 of Appendix C. We compare ILVF-P (no-inference) with the original ILVF-P and MAPPO. The results indicate that ILVF-P (no-inference) consistently underperforms both methods across multiple environments, supporting the effectiveness of the proposed inference mechanism. Detailed results and analysis are provided in Appendix C.

### 5.4 HYPER-PARAMETER ABLATION STUDY

To study the influence of key hyper-parameters on the performance of the ILVF-P algorithm, we conduct a series of ablation studies in the MPE Encircle environment with 20 agents. Specifically, we examine the impact of the initialization strategy of the attention weight matrix $W$, the discretization method of the Gumbel-based attention output $W^g(s)$, and the temperature parameter $\tau$ in the Gumbel-softmax function. Detailed experimental results and analysis are given in Appendix D. Overall, these hyper-parameters exert limited influence on performance, demonstrating the robustness and reliability of the proposed method.

## 6 CONCLUSION

To enhance credit assignment in reward sum settings, we proposed a novel influence-based local value function (ILVF) grounded in A-Q influence, which preserves equivalence to the global value function for policy gradients. A Gumbel-attention mechanism were introduced to learn inter-agent state influence and then infer A-Q influence. Experimental results demonstrated that ILVF consistently improves credit assignment and sample efficiency across various multi-agent environments. In the future, we will consider more general settings such as shared reward, distributed learning, and limited inter-agent communications.

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

# A PROOFS OF THEOREMS

## A.1 PROOF OF THEOREM 1

**Proof.** In a partially observable Markov decision process (POMDP) under the CTDE paradigm, the actor utilizes local observations while the critic has access to the global state. The policy gradient for agent $i$ is given by:

$$\nabla_{\theta_i} J(\theta) = \mathbb{E}_{s \sim p^\pi} \left[ \sum_a \nabla_{\theta_i} \pi(a|o; \theta) Q^\pi(s, a) \right].$$

We decompose the global action-value function $Q^\pi(s, a)$ into an influence-based action-value $\hat{Q}_i^\pi(s, a)$ and a residual term representing the contributions from non-local agents:

$$Q^\pi(s, a) = \hat{Q}_i^\pi(s, a) + \sum_{j \in \mathcal{N} \setminus \mathcal{N}_i^l} Q_j^\pi(s, a),$$

where $\hat{Q}_i^\pi(s, a)$ captures the contributions of agent $i$ and its local neighbors $\mathcal{N}_i^l$. Substituting this into the policy gradient yields:

$$\nabla_{\theta_i} J(\theta) = \mathbb{E}_{s \sim p^\pi} \left[ \sum_a \nabla_{\theta_i} \pi(a|o; \theta) \left( \hat{Q}_i^\pi(s, a) + \sum_{j \in \mathcal{N} \setminus \mathcal{N}_i^l} Q_j^\pi(s, a) \right) \right].$$

To prove the theorem, it suffices to show that the residual term has zero contribution to the gradient:

$$\Phi_i \triangleq \sum_a \nabla_{\theta_i} \pi_\theta(a|o) \sum_{j \in \mathcal{N} \setminus \mathcal{N}_i^l} Q_j^\pi(s, a) = 0.$$

Since the joint policy is factorized as $\pi(a|o) = \prod_{k \in \mathcal{N}} \pi_{\theta_k}(a_k|o_k)$, and $\nabla_{\theta_i}$ only acts on $\pi_{\theta_i}$, we obtain:

$$\nabla_{\theta_i} \pi(a|o; \theta) = (\nabla_{\theta_i} \pi_{\theta_i}(a_i|o_i)) \prod_{k \in \mathcal{N} \setminus \{i\}} \pi_{\theta_k}(a_k|o_k).$$

Let $a = (a_i, a_{-i})$, where $a_{-i}$ denotes the actions of all agents except $i$. Substituting into $\Phi_i$, we have:

$$\Phi_i = \sum_{a_i} \sum_{a_{-i}} \left[ \nabla_{\theta_i} \pi_{\theta_i}(a_i|o_i) \prod_{k \in \mathcal{N} \setminus \{i\}} \pi_{\theta_k}(a_k|o_k) \right] \sum_{j \in \mathcal{N} \setminus \mathcal{N}_i^l} Q_j^\pi(s, a).$$

Rearranging the sums:

$$\Phi_i = \sum_{a_i} \nabla_{\theta_i} \pi_{\theta_i}(a_i|o_i) \sum_{j \in \mathcal{N} \setminus \mathcal{N}_i^l} \sum_{a_{-i}} \left[ \prod_{k \in \mathcal{N} \setminus \{i\}} \pi_{\theta_k}(a_k|o_k) \right] Q_j^\pi(s, a).$$

Since each $Q_j^\pi(s, a)$, for $j \in \mathcal{N} \setminus \mathcal{N}_i^l$, is independent of agent $i$'s action $a_i$ (i.e., $Q_j^\pi(s, a) = Q_j^\pi(s, a_{-i})$), we can write:

$$\Phi_i = \sum_{a_i} \nabla_{\theta_i} \pi_{\theta_i}(a_i|o_i) \sum_{j \in \mathcal{N} \setminus \mathcal{N}_i^l} \sum_{a_{-i}} \left[ \prod_{k \neq i} \pi_{\theta_k}(a_k|o_k) \right] Q_j^\pi(s, a_{-i})$$

$$= \sum_{a_i} \nabla_{\theta_i} \pi_{\theta_i}(a_i|o_i) \sum_{j \in \mathcal{N} \setminus \mathcal{N}_i^l} V_j^\pi(s),$$

where $V_j^\pi(s) = \sum_{a_{-i}} \left[ \prod_{k \neq i} \pi_{\theta_k}(a_k|o_k) \right] Q_j^\pi(s, a_{-i})$ is the state-value function of agent $j$ under state $s$.

Since $\sum_{j \in \mathcal{N} \setminus \mathcal{N}_i^l} V_j^\pi(s)$ is independent of $a_i$ and $\theta_i$, we factor it out:

$$\Phi_i = \left( \sum_{j \in \mathcal{N} \setminus \mathcal{N}_i^l} V_j^\pi(s) \right) \sum_{a_i} \nabla_{\theta_i} \pi_{\theta_i}(a_i|o_i).$$

By the normalization property of the policy, we have $\sum_{a_i} \pi_{\theta_i}(a_i|o_i) = 1$. As a result,

$$\sum_{a_i} \nabla_{\theta_i} \pi_{\theta_i}(a_i|o_i) = 0.$$

Hence,

$$\Phi_i = \left( \sum_{j \in \mathcal{N} \setminus \mathcal{N}_i^l} V_j^\pi(s) \right) \cdot 0 = 0.$$

This implies that

$$\nabla_{\theta_i} J(\theta) = \mathbb{E}_{s \sim p^\pi} \left[ \sum_a \nabla_{\theta_i} \pi(a|o;\theta) \hat{Q}_i^\pi(s,a) \right].$$

The proof is completed. ∎

## A.2 Proof of Theorem 2

Before proving Theorem 2, we give the following lemma.

**Lemma 1.** *Given any policy $\pi(a|s)$, state $s$, it holds that $\dim(\Pi_i^j)) \le |\mathcal{A}_j| - 1$ for any $i, j \in \mathcal{N}$.*

**Proof.** Without loss of generality, let $M \ge N = |\mathcal{A}_j|$, and $s_j'^* = (s_{j1}, ..., s_{jM}) \in \mathcal{S}_j^M$ be a vector consisting of $M$ states from $\mathcal{S}_j$. Define matrix

$$L = \begin{pmatrix} \pi_i(a_{j1}|s) - \pi_i(a_{j1}|s_{\mathcal{N} \setminus j}, s_{j1}') & \cdots & \pi_i(a_{jN}|s) - \pi_i(a_{jN}|s_{\mathcal{N} \setminus j}, s_{j1}') \\ \vdots & \ddots & \vdots \\ \pi_i(a_{j1}|s) - \pi_i(a_{j1}|s_{\mathcal{N} \setminus j}, s_{jM}') & \cdots & \pi_i(a_{jN}|s) - \pi_i(a_{jN}|s_{\mathcal{N} \setminus j}, s_{jM}') \end{pmatrix} \in \mathbb{R}^{M \times N}. \tag{12}$$

Then we have

$$\dim(\Pi_i^j) = \max_{s_j'^* \in \mathcal{S}_j^M} \text{rank}(L). \tag{13}$$

Note that for any $s_{jl}$ with $l \in \{1, ..., M\}$, it holds that

$$\sum_{k=1}^N \left[ \pi_i(a_{jk}|s) - \pi_i(a_{jk}|s_{\mathcal{N} \setminus j}, s_{jl}') \right] = \sum_{a_j \in \mathcal{A}_j} \pi_i(a_j|s) - \sum_{a_j \in \mathcal{A}_j} \pi_i(a_j|s_{\mathcal{N} \setminus j}, s_{jl}') = 0, \tag{14}$$

implying that $L$ is row stochastic. Therefore,

$$\dim(\Pi_i^j)) = \max_{s_j'^* \in \mathcal{S}_j^M} \text{rank}(L) \le |\mathcal{A}_j| - 1, \tag{15}$$

which completes the proof. ∎

**Proof of Theorem 2.** Let $s' = (s_{\mathcal{N} \setminus j}, s_j'), j \in I_i$. From Definition 3, we have:

$$V_i^\pi(s) = V_i^\pi(s_{\mathcal{N} \setminus j}, s_j'), \quad Q_i^\pi(s,a) = Q_i^\pi(s_{\mathcal{N} \setminus j}, s_j', a).$$

Expanding the value function, it holds that:

$$\sum_a \pi(a|s) Q_i^\pi(s,a) = \sum_a \pi(a|s') Q_i^\pi(s,a),$$

which is equivalent to:

$$\sum_{a_j} \left[ \pi_j(a_j|s) - \pi_j(a_j|s') \right] \sum_{a_{\mathcal{N} \setminus j}} \pi(a_{\mathcal{N} \setminus j}|s) Q_i^\pi(s,a) = 0.$$

We denote $\sum_{a_{\mathcal{N}\setminus j}} \pi(a_{\mathcal{N}\setminus j} \mid s) Q_i^\pi(s, a)$ by $Q_i^\pi(s, a_j)$, which yields:

$$\sum_{a_j} \left[ \pi_j(a_j \mid s) - \pi_j(a_j \mid s') \right] Q_i^\pi(s, a_j) = 0.$$

Note that different states $s'$ produce a set of such equations. According to the definition of S-P influence and Lemma 1, there exists a vector $s_j'^* \in \mathcal{S}_j^M$ such that the matrix $L \in \mathbb{R}^{M \times |\mathcal{A}_j|}$ in (12) has $\mathbf{1}_{|\mathcal{A}_j|}$ as the only eigenvector associated with eigenvalue 0, and

$$L \begin{pmatrix} Q_i^\pi(s, a_{j1}) \\ \vdots \\ Q_i^\pi(s, a_{j|A_j|}) \end{pmatrix} = 0. \tag{16}$$

It follows that $Q_i^\pi(s, a_j)$ is constant over $a_j$:

$$Q_i^\pi(s, a_j) = V_i^\pi(s), \forall a_j \in \mathcal{A}_j.$$

Therefore, the action-value function $Q_i^\pi(s, a)$ is independent of $a_j$. ∎

## B  EXPERIMENTAL DETAILS

### B.1  ENVIRONMENT DETAILS

**MPE Encircle.**  Based on the formation control scenario in Agarwal et al. (2019), we design the MPE Encircle environment by introducing fixed observation, transition, and reward coupling structures among agents. In this environment, multiple agents must quickly bypass obstacles and surround a target starting from their initial positions. Specifically, each agent is required to maintain a fixed relative distance to the target and a specific relative position to its neighboring agents. Each agent can observe the states of its neighboring agents as defined by an observation graph.

Since agents are modeled as point-mass particles without physical volume, the state transition of each agent is solely determined by its own state and action. The observation and reward coupling structure is illustrated in Fig. 5. Odd-numbered agents observe the states of their neighboring even-numbered agents, but their rewards depend only on their own states and actions. In contrast, even-numbered agents cannot observe other agents, but their rewards depend on the states of neighboring odd-numbered agents. The reward function for each agent $i$ is defined as:

$$r_i^t = -0.1(d_i - 0.7)^2 - r_{\rho i}^t, \tag{17}$$

$$r_{\rho i}^t = \sum_{j \in \mathcal{N}_i^R} \left( l_{ij} - \frac{2\pi}{n} \right)^2, \tag{18}$$

where $d_i$ denotes the distance between agent $i$ and the target landmark, $\mathcal{N}_i^R$ represents the set of agents with reward coupling to agent $i$, and $l_{ij}$ denotes the angular difference between the relative position vectors of agents $i$ and $j$ with respect to the landmark.

Under this setup, if agents are trained using independent learning, they are able to learn to maintain a fixed distance from the target. However, odd-numbered agents do not learn to maintain specific relative positions with their neighbors since doing so does not bring them any additional rewards. Even-numbered agents attempt to maintain such relative positions, but this effort is ineffective due to their lack of access to neighbors' states. While using a shared global reward could, in theory, lead to globally optimal cooperative strategies, in practice, the credit assignment problem limits such cooperation. Therefore, the MPE Encircle environment serves as a suitable testbed for studying the effectiveness of local value functions.

**Predator Capture Prey (PCP).**  We evaluate the effectiveness of the ILVF-P algorithm in solving heterogeneous multi-agent cooperation problems using the Predator Capture Prey (PCP) environment Seraj et al. (2022). In this environment, *perception agents* and *action agents* must cooperate to successfully capture the prey, and such cooperation relies on the communication topology among agents.

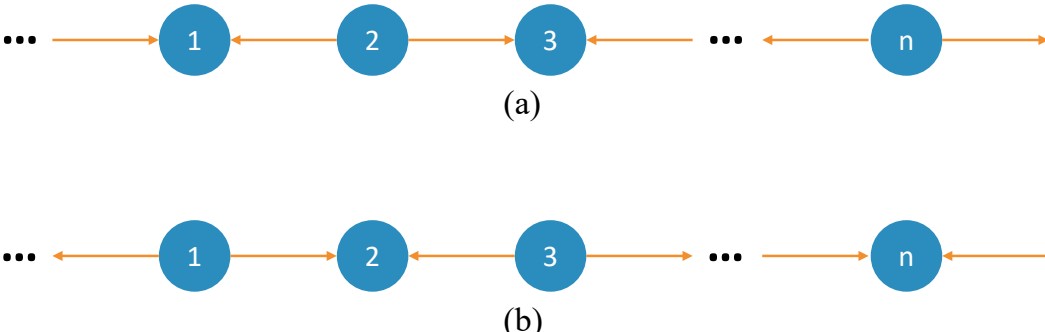

(a)

(b)

Figure 5: Agent Coupling Graph in MPE Encircle.

As illustrated in Fig. 6, agents 1–4 possess perception capabilities but lack the ability to physically capture the prey, while agents 5–8 can execute capture actions but have no perception of the environment. Each perception agent has only local observation capability. Although the underlying state transition coupling among agents theoretically forms a fully connected graph, the actual cooperation pattern is constrained by the communication graph, leading to localized collaboration among agents.

perception agents

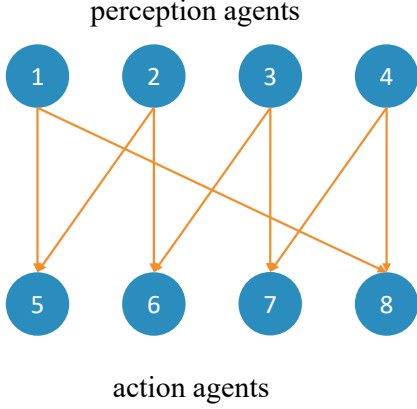

action agents

Figure 6: Observation Topology in PCP.

**Cooperative Adaptive Cruise Control (CACC).** In the CACC environment, vehicles in a platoon aim to transition from arbitrary initial states to a target velocity and spacing while avoiding collisions. Collisions between vehicles result in substantial penalties for all agents, implying that the influence relationships among agents vary across different stages of training. Therefore, this environment is used to investigate the applicability of our method in scenarios with dynamically changing coupling relationships.

**Adaptive Traffic Signal Control (ATSC).** The ATSC environment simulates traffic signal coordination for maximizing network throughput and minimizing delay. It contains two scenarios: *large-grid*, a synthetic grid with bi-directional lanes, and *real-net*, a realistic network built on Monaco's road topology and traffic flow data.

### B.2 BASELINES

We select three classical CTDE-based algorithms—MAPPO, QMIX, and MADDPG—as baseline methods, as they have demonstrated strong performance across a variety of multi-agent environments. The comparison with MAPPO clearly highlights the improvements in sample efficiency and agent cooperation achieved by ILVF. QMIX is included due to its value function factorization mechanism, which also addresses the credit assignment problem under reward sum settings; this allows for a direct comparison that emphasizes the unique advantages of ILVF in such scenarios. MADDPG is chosen

for its effectiveness in partially observable tasks. These comprehensive baselines enables a thorough evaluation of our proposed method's performance and its innovations across diverse scenarios.

Since IPPO has shown effectiveness in SMAC De Witt et al. (2020), we include a variant of IPPO (IPPO-C) where global state is used to train individual value functions, and policies are updated using individual advantages, which ensures a fair comparison with our CTDE-based method. The hyper-parameters related to ILVF-P, MAPPO, IPPO, and the neural networks are summarized in Table 1.

Table 1: PPO and Neural Network Hyper-parameters

| PPO Hyper-parameter | Value | NN Hyper-parameter | Value |
|---|---|---|---|
| ppo_epoch | 15 | hidden_size | 64 |
| use_clipped_value_loss | True | layer_N | 1 |
| clip_param | 0.2 | use_ReLU | True |
| entropy_coef | 0.01 | use_valuenorm | True |
| use_gae | True | use_feature_normalization | True |
| gamma | 0.99 | use_orthogonal | True |
| gae_lambda | 0.95 | gain | 0.01 |
| use_huber_loss | True | lr | 8e-5 |
| huber_delta | 10.0 | opti_eps | 1e-5 |
| eval_episodes | 32 | | |

### B.3 ILVF (NO INFERRING)

The main difference between ILVF (no inferring) and ILVF lies in the design of the critic network. In ILVF (no inferring), critics are required to directly learn inter-agent A-Q influence from the action inputs of individual action-value functions. Therefore, the critic network computes a weighted sum of the action inputs using attention weights perturbed by Gumbel noise, which is then concatenated with the global state to approximate the individual action-value function. For fair comparison, the ILVF (no inferring) still learns individual state-value functions, which are used to construct local value functions and guide policy updates. The architecture of the critic and actor networks with ILVF (no inferring) is illustrated in Figure 7.

The state-value function $V_i^\pi(s)$ and action-value function $Q_i^\pi(s, a)$ are parameterized with neural networks and represented by $V_{\Phi_i^V}^\pi$ and $Q_{\Phi_i^Q}^\pi(s, a)$, respectively:

$$V_{\Phi_i^V}^\pi(s) = f_{\phi_i^V}(s_1, \ldots, s_n), \tag{19}$$

$$Q_{\Phi_i^Q}^\pi(s, a) = f_{\phi_i^Q}\left(s, \sum_{j=1}^n w_{ij}^g a_j\right), \tag{20}$$

where $w_i^g(s) = [w_{i1}^g, \ldots, w_{in}^g]^\top$ are attention weights, with $w_{ij}^g \in [0, 1], \forall i, j \in \{1, \ldots, n\}$. Let $h_i = f_{\phi_i^h}(s) = [h_{i1}, \ldots, h_{in}] \in \mathbb{R}^n, i = 1, \ldots, n$ represent the output vector of the neural network $f_{\phi_i^h}(\cdot)$. The calculation of the attention weights is as follows:

$$w_{ij}^g = \text{sigmoid}(h_{ij} + g_{ij}), \tag{21}$$

where $g_{ij}$ represents a vector of Gumbel noise, $\phi_i = [\phi_i^h, \phi_i^V, \phi_i^Q]$ represents the parameter set of the critic network of agent $i$.

The loss function of the critic network is defined as:

$$\mathcal{L}(\phi_i) = \sum_{t=0}^T \left[(G_i^t - V_{\phi_i}^\pi(s^t))^2 + (G_i^t - Q_{\phi_i}^\pi(s^t, a^t))^2\right], \tag{22}$$

where $G_i^t = \sum_{k=t}^T \gamma^{k-t} r_i^t$ is the discounted return. After the training of each critic network, the state-dependent attention weights with Gumbel noise $w_{ij}^g \in [0, 1], \forall i, j \in \{1, \ldots, n\}$ can be learned, and then are column-wise stacked into a matrix $W^g(s)$, which encapsulates the overall inter-agent A-Q influence.

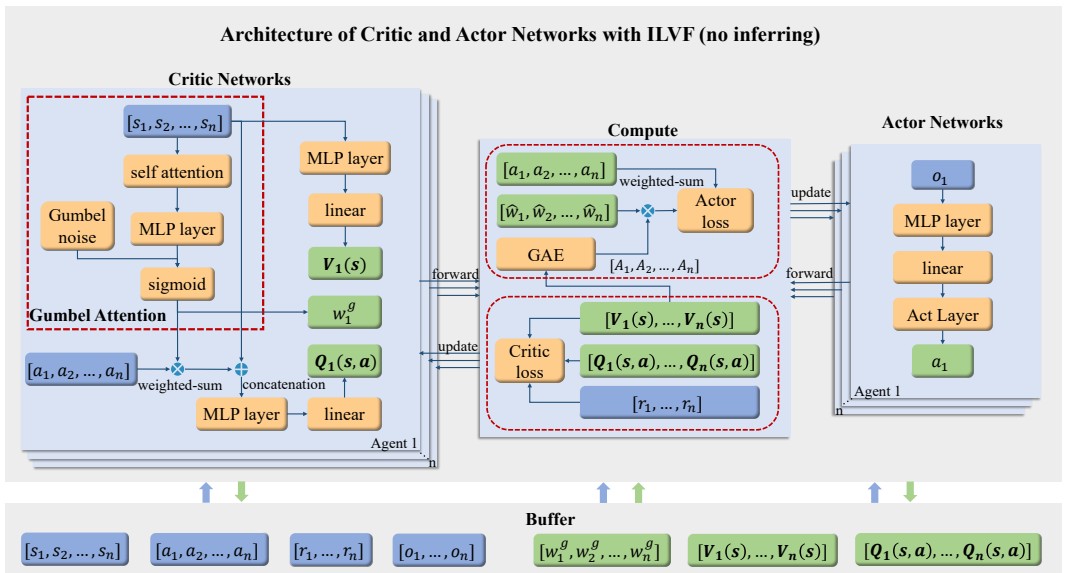

Figure 7: Architecture of critic and actor networks with ILVF (no inferring). The blue boxes represent external inputs, the orange boxes denote modules within the network, and the green boxes indicate intermediate computed variables. The Gumbel attention mechanism is integrated into the critic network to learn the agent-wise A-Q influence relationships, represented by the matrix $W$, which is then used to construct local advantage functions to facilitate policy learning for individual agents.

## C  THE MOTIVATION FOR INFERRING A-Q INFLUENCE

We further conduct experiments to examine the rationale for learning state influence and subsequently inferring A-Q influence. To this end, we design a baseline variant that integrates the Gumbel attention mechanism directly into the action-value function network to learn the A-Q influence among agents directly. We refer to this method as ILVF-P (no inferring), and its architecture is illustrated in Figure 7 of Appendix B.

To ensure a fair comparison, we modify the training framework and network structure of ILVF-P accordingly to derive a consistent setup for ILVF-P (no inferring). We compare ILVF-P (no inferring) against the original ILVF-P and MAPPO algorithms, with the experimental results shown in Figure 8.

From the results, it is evident that ILVF-P (no inferring) significantly underperforms both ILVF-P and MAPPO across multiple environments. This suggests that constructing local advantage functions based on the learned A-Q influence not only fails to enhance policy learning but also yields inferior performance compared to global-value-function-based methods. This outcome implies that the potential influence relationships encoded in the action-value functions are not effectively captured by the learned attention weights.

We hypothesize that the underlying reasons for this result include: (a) inferring A-Q influence from state influence leads to better sample efficiency; and (b) in most multi-agent tasks, the main components of the reward function primarily depend on state variables, while action variables often serve only as control costs or regularization terms. Therefore, the mapping between states and value functions is relatively simpler and easier for neural networks to approximate, whereas the mapping involving actions is more complex and imposes higher demands on model capacity and training stability. While it is possible that ILVF-P (no inferring) could be viable with more sophisticated network architectures, there is currently insufficient evidence supporting its practical applicability.

In summary, inferring A-Q influence from state influence via the Gumbel attention mechanism yields superior performance and greater stability compared to directly learning A-Q influence, making it a more favorable choice.

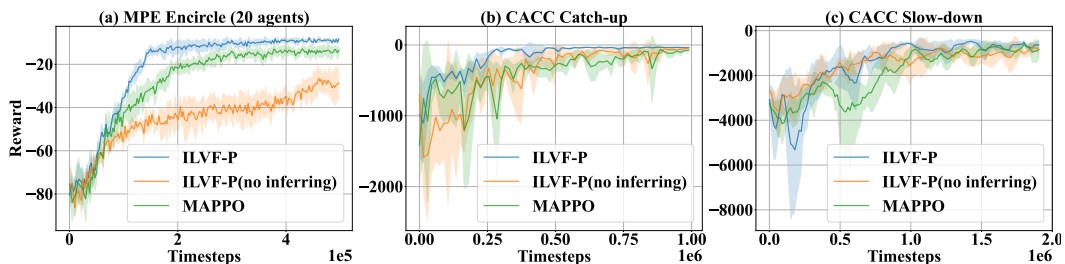

Figure 8: Ablation studies of learning A-Q influence in different scenarios.

## D ABLATION STUDY OF KEY HYPER-PARAMETERS IN THE ILVF FRAMEWORK

To thoroughly investigate the impact of key tunable hyper-parameters on algorithm performance, we design and conduct three groups of ablation experiments in the MPE Encircle (20 agents) environment.

**Initialization of the $W$ Matrix.** We first analyze the effect of the initial values of the influence matrix $W$. In our approach, inter-agent influence relations are dynamically learned during value function training via the Gumbel attention mechanism. Since the policies are continuously updated during training, the learned influence structure is inherently policy-dependent. Thus, different initializations of $W$ may affect policy evolution and, consequently, the final performance.

As shown in Figure 9(a), initializing $W$ as the identity matrix leads to the best convergence speed and final performance. In contrast, initializing $W$ as a full-one matrix slows down convergence and degrades final performance, highlighting the importance of proper credit assignment mechanisms. Further observations suggest that in the MPE Encircle task, a sparser initial $W$ generally yields better performance. This is likely due to the sparsity of inter-agent couplings in this environment. Therefore, further performance improvements may be achieved by tailoring the initialization to the underlying coupling structure. For simplicity and generality, we adopt random initialization as the default setting in this paper.

**Discretization Strategy of $W^g(s)$.** Although the Gumbel attention mechanism outputs a quasi-discrete weight matrix $W^g(s)$, further discretization is necessary. We investigate two strategies: (1) sampling from Bernoulli distributions parameterized by $W^g(s)$; (2) thresholding, where weights below a threshold $c$ are set to 0 and others to 1. The value of $c$ controls the sparsity of the resulting $W$.

Figure 9(b) presents the performance under various discretization methods and thresholds. The results show that using sampling or larger thresholds (e.g., $c = 0.5$ or $c = 0.8$) leads to comparable and superior performance. Lower thresholds cause many weakly contributing local value functions to be discarded, thereby reducing their participation in policy updates and potentially harming overall performance. This supports our hypothesis that selectively removing low-contribution local value functions can improve global performance, even though the idea may seem counterintuitive.

**Temperature Parameter $\tau$ in Gumbel Attention.** We conduct ablation analysis on the temperature parameter $\tau$ of the Gumbel attention mechanism, which directly affects the distribution and sparsity of $W^g(s)$. As shown in Figure 9(c), different values of $\tau$ have limited impact on performance, indicating that the algorithm is robust to this parameter.

**Summary.** In summary, while several key hyper-parameters in the ILVF framework do influence performance, their effects are generally limited. This implies that the proposed method possesses strong stability and generalization capabilities. The superior performance over baseline methods primarily stems from two key innovations: the use of influence-aware local value functions for better credit assignment and policy learning, and the rational design of A-Q influence modeling via state influence inference.

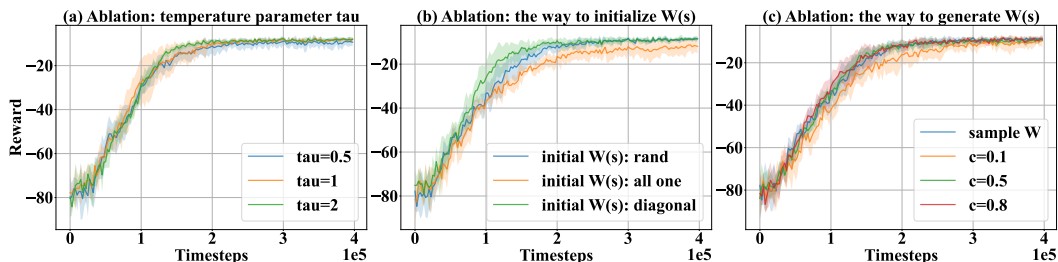

Figure 9: Ablation studies of key hyper-parameters in the ILVF Framework.

## E    LLM USAGE STATEMENT

During the preparation of this manuscript, we employed large language models (LLMs) (OpenAI GPT-4; DeepSeek) as auxiliary tools. The LLM was used to assist in tasks such as improving the clarity of language, polishing grammar, and providing alternative phrasings. It was also occasionally used to generate preliminary drafts of code snippets and to translate technical content. All scientific results presented in this article are our original works, and any texts or codes suggested by the LLM have been carefully reviewed and modified to ensure the accuracy.

