# OpenReview forum: "Dynamic Causal Influence Learning in Cooperative Multi-Agent Reinforcement Learning"
_ICLR.cc/2026/Conference — Submitted to ICLR 2026_

### Official Review · Reviewer_dvUE · 2025-10-20

**Soundness:** 2
**Presentation:** 2
**Contribution:** 2
**Rating:** 4
**Confidence:** 3

**Summary:**

This paper proposes A-Q influence to capture the state-dependent causal influence in MARL for the credit assignment problem. They theoretically prove that each individual ILVF is equivalent to the global value function in terms of policy gradient and infer A-Q influence via state influence. Then the Gumbel attention mechanism is introduced to learn state influence relationships between agents and to construct local advantages for policy gradient. Experiments in MPE demonstrates the effectiveness of ILVF when combined with MAPPO.

**Strengths:**

1.	The method is well motivated by causal influence learning.
2.	The authors provide theoretical definitions and theorems for the proposed method such as the equivalence of global value function and ILVF in terms of policy gradient.

**Weaknesses:**

1.	Several MARL works that utilize causal inference for credit assignment are only discussed in the related works but not compared for a better evaluation of the proposed work.
2.	The influence-based local value function assumes the summation of individual action-value function equals the global action-value function.
3.	There are some typos such as “(CACC ).” At the same time, the format of many citations in this paper is not correct. Moreover, the definition of “span” in Eq (6) is not given.
4.	The method section is a little bit wired and inferring A-Q influence via state influence is not convincing enough as the action information is lost.
5.	The performance improvement of ILVF-P over MAPPO is marginal.

**Questions:**

1.	How does GAE affect the performance of the proposed method?
2.	Is I-IVF in Figure 3 the ILVF-P? Is MQMIX in Figure 3 the QMIX?
3.	Could ILVF be integrated into other MARL algorithms?
4.	How does ILVF perform in other multiagent environments such as SMAC?

---

### Official Review · Reviewer_VcKG · 2025-10-29

**Soundness:** 2
**Presentation:** 2
**Contribution:** 3
**Rating:** 2
**Confidence:** 4

**Summary:**

The paper studies cooperative multi-agent MDPs where the reward exhibits an agent-wise sum structure.
The paper proposes and investigates the notion of A-Q influence for defining local values that preserve the gradient of the global value. As A-Q influence is hard to infer, the paper develops alternative influence notions that are stronger than A-Q but easier to infer in practice.

**Strengths:**

The paper is novel in its study of a hierarchy of inter-agent influences.

**Weaknesses:**

1. There are some soundness/clarity issues:
    - In (1), $\theta$ appears without definition.
    - In Definition 1, it is unclear whether (4) has to hold for any joint action a and for some a.

2. It is unclear what merit Theorem 1 can bring, if the gradient is equivalent before and after introducing the notion of (influence-based )local value functions. In particular, the statement at line 238, “A sparse local value function not only...”, is not well supported.

3. There is no in-depth discussion on Theorem 2 either. It is a sufficient condition for an agent to have no A-Q influence on another. So, is it also a necessary condition and, if not, how strong is the condition?

4. It is unclear what role the reward-sum structure is playing in this paper:
    - Do we have an ablation study where we ignore the  reward-sum structure by pretending all agents’ reward is equal to the total reward?
    - In Figure 3, what curves are exploiting and ignoring the reward-sum structure?

5. The method developed in Section 4.3.1 seems loosely connected with the theoretical results in Section 4.2. In particular, the training loss for the attention weights is not related to the notions of S-V or S-P. The statement at 341-342 is therefore not well supported.

**Questions:**

All my concerns/questions are in the Weaknesses section.

---

### Official Review · Reviewer_QRYP · 2025-11-01

**Soundness:** 1
**Presentation:** 2
**Contribution:** 1
**Rating:** 2
**Confidence:** 4

**Summary:**

This paper introduces the concept of influences between actions and value functions and aims to incorporate these influences into value decomposition.

**Strengths:**

The definition and use of the influences of actions on value functions seem novel and interesting.

**Weaknesses:**

- Confusion in the setting: This paper aims to tackle the multi-agent credit assignment problem, which is usually considered in cooperative settings. In Section 3.1, the authors assume that the team reward is the sum of individual rewards and build individual value functions based on the individual rewards. However, individual rewards are usually assumed to be unavailable in the credit assignment problem, and the main issue lies in determining the contributions of agents to the team reward. In the theoretical results section, it seems that the authors assume individual rewards are provided, but they only consider the team reward in the proposed method. The exact setting the authors are considering is therefore confusing.

- Disconnection between theory and method: This point is related to (1) above. Although the definitions of influences are interesting, the proposed method shown in Figure 2 does not clearly incorporate the influences as defined. It appears that the proposed method is primarily an architectural design for representing states and actions with local and global value functions, using an attention mechanism to implicitly learn influences without any guarantee. I expected the authors to compute the influences explicitly and then leverage them for credit assignment, but the proposed method seems to be just an instance of a mixing network, as commonly seen in the value decomposition literature.

- Unimpressive evaluations: In the main results shown in Figure 3, the performance improvement is marginal. I recommend that the authors test their method in more challenging environments that require explicit credit assignment.

Minor Comments

- References appear to be formatted incorrectly, e.g., Kiran et al. (2021); Chen et al. (2024)  ; Wang et al. (2023a); Tang et al. (2025) should be (Kiran et al. (2021); Chen et al. (2024); Wang et al. (2023a); Tang et al. (2025))

- r_i^t+1 should be r_i^t in Figure 1.

**Questions:**

See Weaknesses

---

### Official Review · Reviewer_YahM · 2025-11-01

**Soundness:** 3
**Presentation:** 3
**Contribution:** 2
**Rating:** 4
**Confidence:** 4

**Summary:**

This paper proposes a novel Influence-based Local Value Function (ILVF) method to resolve the Credit Assignment problem in Cooperative Multi-Agent Reinforcement Learning (MARL). The authors first define A-Q influence to capture the state-dependent causal influence relationship between individual actions and individual action-value functions. Theoretically, the authors prove that the ILVF constructed from this influence is equivalent to the global value function for policy gradient estimation (Theorem 1). To efficiently learn the dynamic A-Q influence, the paper proposes inferring A-Q influence by learning State Influence (S-V and S-P influence), where state influence is learned via a Gumbel Attention Mechanism.

**Strengths:**

1.The core contribution is the theoretical proof that the proposed ILVF is equivalent to the global value function in terms of policy gradient (Theorem 1). This provides strong theoretical support for using a local value function to guide policy updates, avoiding the common monotonicity constraints or compounded approximation errors often faced by factorization methods (e.g., QMIX).
2.The author's experiment is relatively sufficient. ILVF-P consistently shows faster convergence speed and higher final returns across multiple challenging multi-agent environments, proving its effectiveness in improving credit assignment.

**Weaknesses:**

1. The author proves (Theorem 2) that if an agent does not possess S-V influence, then it also does not possess A-Q influence. However, the author does not prove that possessing S-V influence automatically grants A-Q influence. Nevertheless, the author argues that A-Q influence refers to the causal impact of one agent on another.
2. The authors used the Gumbel attention mechanism to learn the state influence relationships among agents; however, the influence relationship learned in this way is still a correlation rather than a causal relationship.
3. The ablation study in Section 5.3 shows poor performance for ILVF-P (no inferring). The authors hypothesize that this is because the reward function primarily depends on states, with action influence mediated by environmental dynamics. While plausible, the paper lacks a deeper technical analysis or theoretical argument for why direct A-Q influence learning fails so significantly. A more in-depth analysis would better help readers understand the necessity of the "Inferring" mechanism.
4. Experiments cover diverse environments (MPE, PCP, CACC, ATSC), but the authors should validate the effectiveness of their methods in typical multi-agent scenarios such as SMAC and GRF.

**Questions:**

1. Will the Gumbel attention mechanism introduce additional complexity? A discussion or analysis of the computational overhead compared to baselines would be valuable.
2. Please explain the specific motivation for converting the continuous output $W^g(s)$ from the Gumbel noise into the sparse binary matrix $\hat{W}$ in Section 4.3.2. Ablation study D.2 compares Bernoulli sampling and thresholding. Please briefly state the final choice of discretization method (which one is the default?) in the method section, and explain why this sparsity helps credit assignment more than using continuous weights (i.e., the benefit of sparsity itself).

---

### Meta-Review · Area_Chair_ssNu · 2026-01-05

**Summary:**

This paper addresses credit assignment in cooperative multi-agent reinforcement learning by introducing A-Q influence—a state-dependent measure of causal influence between agents' actions and value functions. The authors propose influence-based local value functions (ILVFs), prove their equivalence to global value functions for policy gradients (Theorem 1), and develop a Gumbel attention mechanism to learn state influence for inferring A-Q relationships. The method is integrated with MAPPO and evaluated across several environments.

The reviewers acknowledge the novelty of the influence-based framework and the theoretical contributions. However, significant concerns emerged regarding the gap between theory and practice, the realism of assumptions, and the strength of empirical validation.

**Reviewer Concerns:**

Outstanding concerns:
- Theory-method disconnect (QRYP, VcKG): The Gumbel attention mechanism learns correlations rather than the defined causal influences, with no explicit guarantee connecting training objectives to S-V/S-P definitions
- Problematic assumptions (QRYP): Individual rewards are assumed available, which conflicts with standard credit assignment settings where only team rewards exist
- Marginal empirical improvements (all reviewers): Performance gains over MAPPO are modest across environments
- Missing standard benchmarks (YahM, dvUE): No experiments on SMAC or GRF, which are standard MARL testbeds
- Insufficient analysis (YahM, VcKG): Why direct A-Q learning fails and the necessity of the inference mechanism lack rigorous justification

**Reviewer Scores:**

The reviewers would not have changed the score since there was no rebuttal.

---

### Decision · Program_Chairs · 2026-01-26

Reject